# Molecular Mechanisms of Scombroid Food Poisoning

**DOI:** 10.3390/ijms24010809

**Published:** 2023-01-03

**Authors:** Yury V. Zhernov, Mark Y. Simanduyev, Olga K. Zaostrovtseva, Ekaterina E. Semeniako, Kseniia I. Kolykhalova, Inna A. Fadeeva, Maria I. Kashutina, Sonya O. Vysochanskaya, Elena V. Belova, Denis V. Shcherbakov, Vitaly A. Sukhov, Ekaterina A. Sidorova, Oleg V. Mitrokhin

**Affiliations:** 1Department of General Hygiene, F. Erismann Institute of Public Health, I.M. Sechenov First Moscow State Medical University (Sechenov University), 119435 Moscow, Russia; 2Department of Chemistry, Lomonosov Moscow State University, 119991 Moscow, Russia; 3Center of Life Sciences, Skolkovo Institute of Science and Technology, 121205 Moscow, Russia; 4Center for Medical Anthropology, N.N. Miklukho-Maclay Institute of Ethnology and Anthropology, Russian Academy of Sciences, 119017 Moscow, Russia; 5Department of Medical and Biological Disciplines, Reaviz Medical University, 107564 Moscow, Russia; 6The Baku Branch, I.M. Sechenov First Moscow State Medical University (Sechenov University), Baku AZ1141, Azerbaijan; 7Department of Foreign Language, Faculty of World Economy, Diplomatic Academy of the Russian Foreign Ministry, 119034 Moscow, Russia; 8Department of Public Administration in Foreign Policy, Diplomatic Academy of the Russian Foreign Ministry, 119034 Moscow, Russia; 9Loginov Moscow Clinical Scientific and Practical Center, 111123 Moscow, Russia; 10Department of Public Health Promotion, National Research Centre for Therapy and Preventive Medicine, 101990 Moscow, Russia; 11Department of Therapy, Clinical Pharmacology and Emergency Medicine, A.I. Yevdokimov Moscow State University of Medicine and Dentistry, 127473 Moscow, Russia

**Keywords:** food hypersensitivity, pseudo-allergy, mast cells, non-immunologic food intolerance, scombroid food poisoning, scombrotoxicosis

## Abstract

Scombroid food poisoning (SFP) is a foodborne disease that develops after consumption of fresh fish and, rarely, seafood that has fine organoleptic characteristics but contains a large amount of exogenous histamine. SFP, like other food pseudo-allergic reactions (FPA), is a disorder that is clinically identical to allergic reactions type I, but there are many differences in their pathogenesis. To date, SFP has been widespread throughout the world and is an urgent problem, although exact epidemiological data on incidence varies greatly. The need to distinguish SFP from true IgE-associated allergy to fish and seafood is one of the most difficult examples of the differential diagnosis of allergic conditions. The most important difference is the absence of an IgE response in SFP. The pathogenesis of SFP includes a complex system of interactions between the body and chemical triggers such as exogenous histamine, other biogenic amines, cis-urocanic acid, salicylates, and other histamine liberators. Because of the wide range of molecular pathways involved in this process, it is critical to understand their differences. This may help predict and prevent poor outcomes in patients and contribute to the development of adequate hygienic rules and regulations for seafood product safety. Despite the vast and lengthy history of research on SFP mechanisms, there are still many blank spots in our understanding of this condition. The goals of this review are to differentiate various molecular mechanisms of SFP and describe methods of hygienic regulation of some biogenic amines that influence the concentration of histamine in the human body and play an important role in the mechanism of SFP.

## 1. Introduction

Food pseudo-allergic reactions (FPA), also known as false food allergies (FFA), are one type of food hypersensitivity under which a pathological process is clinically similar to food allergic reaction type I, but has no immunological stage [1,2]. Currently, non-immunological adverse reactions to food, including FPA, are prevalent in 15–20% of the population, which is four times as much as true food allergies [3]. A special type of FPA is scombroid food poisoning (SFP), also known as scombrotoxicosis, scombrotoxism, scombroid ichthyotoxicosis, or Mahi-Mahi flush [4]. To date, SFP has been widespread throughout the world and is an urgent problem, although exact epidemiological data on incidence varies greatly. SFP is considered to be responsible for approximately 40% of all food intoxications in the EU and the US [5]. The incidence ranges from 2–5 cases per million people in New Zealand, France, and Denmark to 31 cases per million people in Hawaii [5]. Despite the prevalence of SFP, healthcare professionals’ awareness of this disease is low. Thus, according to a prospective descriptive cross-sectional study in Tanzania, the awareness of healthcare professionals about SFP was 60%, of which only 6.3% had good knowledge [6].

The clinical presentation of SFP is similar to that of an IgE-associated food allergy reaction. It may include hives, angioneurotic oedema, gastrointestinal symptoms, bronchospasm, skin redness, headache, oedema, hypotension, and shock [7,8]. Cases are known when such an FPA reaction as SFP leads to severe intoxication and death [9,10,11,12,13]. Therefore, for correct SFP diagnosis, it is important to evaluate the patient’s previous allergic anamnesis and indirect epidemiologic signs, and to perform a molecular allergology examination, such as ImmunoCAP (Immuno Solid-phase Allergy Chip) ISAC assay, MeDALL allergen-chip assay, or highly sensitive ELISA-based assay, which will help rule out a true IgE-associated allergy to fish and seafood associated with sensitization to parvalbumins [14,15,16,17].

There are no antigen-specific immune complexes in the SFP molecular mechanism, but there is histamine release, complement activation, atypical eicosanoids synthesis, and inhibition of bradykinin decomposition. The basis of SFP is the non-specific release of the inflammatory mediators from the target cells when mast cells are affected directly by the food antigens and indirectly when the complement, kinine, and other biological systems induce the clinical symptoms of allergy [18,19,20]. However, the molecular mechanisms of this disease are still poorly understood.

This review aims to provide a modern understanding of the molecular mechanisms of FPA reactions as well as a discussion of recent discoveries in the mechanism of formation of SFP, which can be considered on the one hand as one of the variants of food hypersensitivity with a predominance of pseudo-allergic reactions, and on the other hand as a typical food intoxication. Understanding the molecular mechanisms of food pseudo-allergic reactions, such as SFP, is important for differential diagnosis and personalized treatment of such conditions, as well as improving the quality of life of patients.

## 2. Contributing Factors of Scombroid Food Poisoning

SFP is a foodborne disease that develops after consumption of fresh fish and, rarely, seafood that has fine organoleptic characteristics but contains a large amount of exogenous histamine. According to statistics, exogenous histamine concentration in the blood of 1–2 ng/mL leads to increased gastric secretion and heart rate; 3–5 ng/mL causes headache, itching, and tachycardia; 6–8 ng/mL lowers blood pressure; 7–12 ng/mL induces bronchospasm, and concentration over 100 ng/mL leads to heart arrest [21]. To evaluate the mean histamine concentration in fish that leads to poisoning in 2017, Columbo et al. conducted a review with an analysis of the cases of histamine poisoning and came to the conclusion that SFP occurs with a mean concentration of 1100 mg/kg and that it varies from 423 to 2900 mg/kg [22]. Obviously, such significant dispersion may be caused by a large number of mechanisms, from exogenous histamine intoxication and histamine liberators’ development to the individual activity of DAO of different patients. 

The SFP mechanism can be influenced by a combination of three external contributing factors (fish species, saprophytic microorganism species that convert histidine to histamine, and environmental conditions), as well as internal individual factors such as exogenous histamine consumed, inactivation speed, and individual sensitivity [23]. The totality of this can be called the SFP formula (Figure 1). 

### 2.1. Fish Species

The Codex Alimentarius and the European Commission designate Clupeidae (herring, sardines, shad), Coryphaenidae (mahi-mahi), Engraulidae (anchovies), Pomatomidae (bluefish), Scombridae (mackerel, tuna, scomber, skipjack, bonito), and Scomberesocidae (saury) as potentially dangerous sources of SFP [24,25]. There had been a lengthy discussion on whether Salmonidae are scombrotoxic families until the Food and Agriculture Organization of the United Nations (FAO) 2018 review concluded that Salmonidae do not present such a danger as the families listed above, but they may participate in the development of SFP if consumed after the expiry date [26]. The most common sources of SFP among Scombridae are tuna, mackerel, and scomber [27]. *Istiophorus* sp. (sailfish) and *Seriola lalandi* (yellowtail kingfish) are among other potentially scombrotoxic species [28,29].

Exogenous histamine is a product of muscle tissue histidine decarboxylation that occurs in these fish. Because high levels of histamine may form in fish with fine organoleptic characteristics, fresh fish may contain histamine as well as spoiled fish [30]. However, histamine intoxication most often develops after ingestion of spoiled fish, especially if it is from the *Scombridae* family. It occurs as a result of bacteria contaminating fish fermenting for 6 h at temperatures greater than 25 °C or longer if the temperature is lower [31]. These microorganisms are found in various products either as normal or contaminating flora.

The sources of exogenous histidine in the tissue can be not only fish but also crabs. There have already been several cases of SFP due to the consumption of crabs, such as mangrove crab *Scylla serrata* [12]. Cases of SFP resulting from consumption of dolphins (Delphinidae) have also been described [32].

### 2.2. Bacteria Species

Histidine from different fish species can be found in food products initially or may be naturally released as a result of proteolysis induced by the processing of food or storage conditions [33]. The L-histidine decarboxylase found in *E. coli*, *Proteus*, *Klebsiella*, and *Morganella morganii* [34] turns the histidine of fish tissues into histamine [35]. All *Morganella morganii* strains produce histamine in quantities of more than 5000 ppm. It was shown that one of these strains produced 5253 ppm in tuna that was stored at 25 °C and 2769 ppm at 15 °C [30]. It was also reported that *Enterobacter aerogenes* and *Raoultella planticola* participate in SFP [36]. Significant histamine production was found in five species of photobacteria: *Photobacterium angustum*, *Photobacterium aquimaris*, *Photobacterium kishitanii*, *Photobacterium damselae* and *Photobacterium phosphoreum*. Of all of them, only *Photobacterium phosphoreum* was associated with the SFP outbreak. However, the classification of this microorganism has been changed and now it is identified either as *Photobacterium kishitanii* or *Photobacterium aquimaris*. Because of that, researchers cannot clearly tell whether these bacteria are capable of histamine production or not, but Bjornsdottir-Butler et al. have found out that, although these microorganisms have decarboxylase and they produce histamine, their decarboxylase genes are different from the ones of *E. coli* [34]. *Photobacterium phosphoreum* was capable of synthesizing histamine at a temperature lower than 10 °C [30]. 

Table 1 shows the bacteria species and food storage conditions that lead to the accumulation of exogenous histamine in fish and, as a result, make them dangerous to eat.

### 2.3. Environmental Conditions

In 2020, the FDA determined that a histamine concentration of 500 mg/kg is sufficient to cause SFP. However, evaluating the exogenous histamine concentration in a shipment of fish is difficult because the distribution of histamine in a fish is not homogenous and often depends on environmental conditions. Therefore, the FDA assumes that if any part of one fish or of the whole shipment contains more than 500 mg/kg of histamine, then it is most likely that other parts may contain concentrations higher than 500 mg/kg [45]. In the European Union, fish species with high levels of histidine are monitored in a special way. The maximum allowed level of histamine in fish of the “risk group” is 100 mg/kg [46].

Fish storage and cooking temperatures are important environmental factors in the pathogenesis of SFP. Keeping products at a cold temperature prevents the growth of the bacteria and the synthesis of their histidine decarboxylase [47]. Despite the fact that the Food and Drug Administration USA (FDA) introduced regulations on the rapid cooling of caught fish to ensure the safety of seafood, SFP remains one of the most prevalent diseases in the world associated with fish [48]. Histamine formation may occur in fish that have been stored or transported incorrectly [30,49]. For example, keeping fish above 4.4 °C for more than 4 h after initial chilling will result in the accumulation of free histidine in the fish [50]. Chung B.Y. et al. found that cooking (grilling and frying) causes the inactivation of histamine-producing bacteria, but histamine can remain in cooked foods. Increasing the cooking temperature can increase histidine decarboxylase activity and, therefore, histamine production. When heated above 50 °C, histidine decarboxylase is inactivated, but remaining exogenous histamine can still lead to SFP [51].

Another factor influencing the formation of endogenous histamine is sodium chloride, which is used in cooking in the form of sea salt and table salt. Histamine production may also be stopped by salt, which is a histidine decarboxylase inhibitor. Research shows that when salting fish, there are no signs of microbiological growth or elevation in the concentration of the amines [47]. Still, some bacteria can produce biogenic amines even in a 12 percent salted environment. For example, an increased concentration of biogenic amines was noticed in salted sardines. Histamine concentrations are also elevated in marinated fish, even if it is stored properly. Biogenic amine levels do not decrease even after sterilization; studies show that up to 90% of amines remain in the products even after this procedure [52].

Humidity is also an environmental factor that can influence the formation of exogenous histamine, so the concentration of endogenous histamine in fish, in terms of the total weight of the product, can increase its drying or humidity loss during cooking [51]. Maintaining anoxic conditions, such as when vacuum-packing fish, is not an effective method of preventing the formation of histamine [50].

Therefore, the quality of fresh fish is determined by the proper timing of cooling, whereas the quality of canned fish is determined by multiple factors, such as the freshness of the primary products, fish species that are canned, the method of processing of the fish, and conditions of storage and transportation.

## 3. Molecular Mechanisms of Scombroid Food Poisoning

Since SFP is an FPA, exogenous histamine intoxication is strictly dose dependent. Increased intoxication with exogenous histamine leads to an increase in symptoms and a deterioration in the human condition. The need to distinguish SFP from true IgE-associated allergies to fish and seafood is one of the most difficult examples of the differential diagnosis of allergic conditions. There are several key differences in the mechanisms between SFP and true IgE-associated allergies to fish and seafood (Table 2).

True fish allergy is usually associated with sensitization to parvalbumins, which are physically and chemically resistant. Parvalbumins are found in various fish species and their cross-activity is often observed, so it is not enough to avoid only certain fish species to prevent the allergic reaction [53]. It is important to note that parvalbumins may be inhaled, as well as food allergens. That is why allergic reactions to fish have severe clinical courses not only when fish is ingested but also when the smell of fish is inhaled. The reaction in this case is reproducible and may develop even after the shortest contact with fish [54]. The main factors in the pathogenesis of true allergy to fish and seafood involve not only immune complexes but also the stage of silent sensibilization to the allergen. After that, the allergic reaction may be triggered by the smallest amount of allergen and develop more slowly than in SFP. The target cells are activated directly by IgE [55].

Meanwhile, SFP is induced by the presence of saprophytic bacteria and an increased amount of exogenous histamine and histamine-like chemicals in fish caused by improper storage [56]. The intensity of the SFP reaction is directly proportional to the amount of exogenous activator consumed, and the reaction itself develops rapidly.

Below, we consider some of the known molecular mechanisms of SFP, the role of which has been proven to varying degrees.

### 3.1. The Role of Exogenous Histamine and Its Metabolic Pathways

Endogenous histamine, contained in the mast cells and basophiles, is one of the most important mediators of non-IgE-mediated clinical reactions, as well as IgE-mediated ones. There are two known ways of histamine metabolism in the human body: methylation by histamine-N-methyltransferase (HNMT) and oxidative degradation by diamine oxidase (DAO) [57]. Besides the endogenous histamine synthesized in the human body, this substance can enter exogenously via the alimentary route and cause SFP in high doses. Large doses of exogenous histamine can cause poisoning: 100 mg and up can cause light intoxication, while 1000 mg and up can cause severe intoxication [58].

The source of exogenous histamine is often fish, cheese, and dairy products contaminated by saprophytic bacteria. According to the FDA, high quality fish must contain histamine at a concentration of less than 10 parts per million, and the level of defect action must be 50 parts per million [59]. While histamine levels of ≥50 parts per million in fish tissues are considered to be a sign of decay, levels of ≥500 parts per million indicate a potential threat to human health and regulatory measures are taken regarding these products [34]. In concentrations greater than 500 parts per million, generalized reactions were observed [60]. Another food product besides fish in which exogenous histamine can be formed under the action of saprophytic microflora is cheese and dairy products. It is known that *Lactobacillus buchneri*, which possesses histidine decarboxylase and plays an important role in the dairy industry, is capable of converting histidine into histamine. Potentially, it may be responsible for outbreaks of histamine poisoning with different sorts of cheese. The toxicity of this histamine may probably double because of other biogenic amines that are produced by the bacteria in cheese. Some of these amines inhibit histamine-metabolizing enzymes and elevate the concentration of exogenous histamine, significantly slowing its decomposition [61].

Many of the toxic effects of exogenous histamine in SFP are realized through endogenous histamine receptors. These receptors are H1R, H2R, H3R, and H4R, and they all belong to the family of G-protein-coupled receptors [57]. The sarcolemma of vascular smooth myocytes contains H2 histamine receptors that are associated with GS protein. Upon reacting with the H2 receptors, exogenous histamine triggers an intracellular biochemical cascade, as follows in Figure 2.

In the resting state, a GDP molecule is bound to the α_s_ subunit of the G_s_ protein, keeping it in an inactive conformation. However, the interaction of histamine with the receptor contributes to a change in the conformation of the a_s_ subunit, which leads to a decrease in its affinity for GDP and an increase in its affinity for GTP (i.e., GDP is detached from the α_s_ subunit and GTP is attached in its place). The point is that the binding of the α_s_ subunit to GDP was important to keep this subunit in the complex with the β and γ subunits, but now that it is bound to GTP, the αS subunit changes its conformation a second time and disconnects from the β and γ subunits. Then, the α_s_ subunit uses lateral diffusion (lateral movement along the inner surface of the cell plasma membrane) to the transmembrane enzyme adenylyl cyclase. Once attached to adenylyl cyclase, the α_s_ subunit activates it. Once activated, adenylyl cyclase begins to convert an ATP molecule into cAMP. In turn, four molecules of cAMP bind to the active centers of two regulatory subunits of type 1 PKA enzyme located in the sarcoplasm of smooth myocytes. As a result, type 1 PKA changes its conformation and its two catalytic subunits are activated and begin phosphorylation of serine and threonine OH-groups of proteins [57,62].

In this case, the target of type 1 PKA is MLCK (myosin light chain kinase), which it inactivates by phosphorylation. The fact is that in the resting state, MLCK phosphorylates myosin regulatory light chains and thereby stimulates the activation of the ATP-ase domain of myosin heads, leading to ATP hydrolysis and, as a consequence, contraction of smooth myocytes. But as we mentioned earlier, the cascade of reactions triggered by histamine results in inactivation of MLCK and, as a consequence, relaxation of vascular smooth myocytes, which in turn leads to vasodilation [63].

H2 receptors are also located on the sarcolemma of typical and atypical cardiomyocytes. When activated, PKA begins to phosphorylate L-type Ca channels, promoting their opening and, as a consequence, the entry of large amounts of Ca from the extracellular space into the intracellular on a concentration gradient. In addition, PKA also phosphorylates RYR2 (Ryanadine receptor 2) located on the membrane of the sarcoplasmic reticulum, promoting their opening, which will lead to the release of Ca from the SR lumen into the cell cytosol. All this will lead to an increase in Ca concentration in the sarcoplasm of atypical cardiomyocytes and, as a consequence, to an increase in HR [62,64]. 

It is worth noting that in the sarcoplasm of typical cardiomyocytes there is a phospholamban protein that binds to SERCA (Sarco/endoplasmic reticulum Ca—ATPase) at rest and reduces its affinity for Ca, thus preventing Ca entry back from the cardiomyocyte sarcoplasm into the lumen of the sarcoplasmic reticulum. However, PKA phosphorylates phospholamban by the OH groups of serine and threonine and thereby promotes its uncoupling from SERCA. This creates an opportunity for Ca to leave the sarcoplasm and allows the muscle to relax for even more accelerated contraction [62,65,66].

Vascular epithelial cells have H1 histamine receptors that are associated with the G_q_ protein. Upon reacting with H1 receptors, histamine triggers an intracellular biochemical cascade as follows in Figure 3.

The α_q_ subunit of the G_q_ protein undergoes similar changes as the α_s_ subunit of the G_s_ protein. In this case, the α_q_ subunit activates the enzyme PLC (Phospholipase C), which is located on the inner surface of the endotheliocyte membrane. PLC begins to hydrolyze PIP_2_ (phosphotidylinositol diphosphate), which is a typical representative of cell membrane phospholipids, to inositol triphosphate (phospholipid head) and diacylglycerol (phospholipid tail). DAG remains in the cell membrane and takes part in PKC activation. IP_3_ binds to IP_3_ receptors located on the ER membrane and stimulates them. It is important to note that IP_3_ receptors are Ca channels and that after interaction with IP_3_ they open, promoting the release of large amounts of Ca ions from the ER into the cell cytosol along a concentration gradient. Ca will be involved in PKC activation, but most importantly, four Ca ions bind to calmodulin located in the cytosol of endotheliocyte, forming a calmodulin-4Ca complex [62,67].

The endothelial cell cytosol contains the enzyme eNOS (endothelial NO synthase), which is normally in the inactive conformation. Calmodulin-4Ca complex binds to eNOS and converts it to the active conformation. Once activated, eNOS catalyzes the synthesis of NO from the amino acid L-arginine. Next, NO synthesized by endothelial cells enters the sarcoplasm of vascular smooth myocytes by simple diffusion, where it activates sGC (soluble guanylate cyclase)—its feature is that it is located in the sarcoplasm of myocytes, not on the sarcolemma. When activated, sGC catalyzes the conversion of GTP to cGMP. In turn, two molecules of cGMP bind to PKG and activate it. PKG begins to phosphorylate MLCK by serine/threonine OH-group, inactivating it, which leads to relaxation of vascular smooth myocytes and, consequently, to vasodilation [68,69].

In addition, Ca, which enters the cytosol of endotheliocytes in large quantities under the action of IP_3_, stimulates actomyosin interaction in the cytoskeleton, which in turn leads to the contraction of endothelial cells and the formation of interendothelial gaps between them. By this mechanism, there is an increase in capillary permeability in SFP [70].

### 3.2. The Role of Other Biogenic Amines and Their Metabolic Pathways

In studies regarding the pathological mechanism of SFP, researchers’ attention was focused on the reactions caused by the products’ high histamine content. However, it should be taken into account that other biogenic amines, known as putrefactive (cadaverine, tryptamine, tyramine, serotonin, and others) and/or polyamines (putrescine, spermine, and others), are also capable of inducing similar reactions by influencing the metabolism of histamine in the human body [71].

Tyramine was the first discovered substrate of monoamine oxidase (MAO) and was derived from cheese. Tyramine oxidase is found in a high concentration in the mucosa of the intestines. As a result of bacterial decarboxylation of amino acids, vasopressor amines are formed [72]. Besides its indirect influence on histamine, tyramine also plays a role in adverse reactions of MAO inhibitors. These reactions include headaches and hypertensive crises [73]. Hypertensive crises caused by consuming products containing tyramine and other biogenic amines have also been reported after consuming wine, beer, some fruit and vegetables (sauerkraut, banana peels, avocado, beans) [74,75], and fish [76]. Tyramine, putrescine, and other biogenic amines are found in fish and are produced by bacteria by the decarboxylation of homologous free amino acids [77]. It should be noted that the amount of tyramine and spermine increases when the fish is half-dried [78]. Thus, tyramine and other biogenic amines play an important role in SFP, especially when associated with semi-dried fish.

Other biogenic amines that can potentiate SFP are tryptamine and phenylethylamine. Tryptamine may inhibit DAO, and phenylethylamine inhibits both DAO and HNMT. Although only tyramine may reach concentrations high enough in fish to be toxicologically significant, the synergetic action of all these amines may cause a SFP reaction while eating fish [79].

Also, numerous studies have been conducted to identify the amines which are produced by the *Lactobacillus buchneri* colonizing cheese. Such amines as histamine, cadaverine, putrescine, tryptamine, and phenylethylamine are found in many sorts of cheese [21]. Experiments with *E. faecalis* EF37 have shown that increasing the temperature from 16 to 44 °C is correlated with a more rapid and intensive accumulation of tyramine [22]. In 2015, Bargossi et al. investigated the temperature most favorable for tyrosine decarboxylase derived from *E. faecalis* and found out that it functioned most intensively in the temperature range from 30 to 37 °C [80,81].

Tyramine acts by the mechanism as follows in Figure 4.

There is the NET (Norepinephrine transporter) on the membrane of the synaptic bulb of adrenergic neurons. Tyramine from the synaptic cleft is transported to the synaptic bulb by the NET (Norepinephrine transporter) in exchange for NE itself. Once inside the synaptic bulb, tyramine is transferred inside the vesicle by the vMAT (vesicular monoamine transporter) located on the membrane of these vesicles, in exchange for NE. This NE then enters the synaptic cleft again via the NET in exchange for tyramine and the cycle repeats. As a result, there is a sharp increase in the concentration of NE in the synaptic cleft [82]. According to the mechanism of tyramine action, it should be assumed that tyramine, by increasing NE concentration and vasoconstriction, may mitigate the systemic effects of histamine, which is the main BAS responsible for the development of SFP.

### 3.3. The Role of the Complement System

In many cases of SFP, there is an activation of the complement system, and they are extremely clinically significant because complement activation-related pseudo-allergy (CARPA) has an unpredictable course and may result in a lethal outcome [83].

The complement system participates in natural immune response, but it may also be involved in SFP pathogenesis and damage the cells with anaphylatoxins: C3a, C5, and the membrane attack complex (C5b-C9). There are nanoparticles that can activate the complement without IgE but with anaphylatoxins C3a and C5a that bind with the mast cells and induce the release of various vasoactive mediators that lead to the clinical presentation of the type 1 hypersensitivity reaction, but actually it is CARPA. These nanoparticles are found in some drugs and food products [84,85].

The complement cascade in CARPA may be realized in different pathways: the classic pathway (CP), the lectin pathway (LP), or the alternative pathway (AP) (Figure 5) [20].

#### 3.3.1. Classic Pathway

Upon activation, C1 component of complement cleaves C4 into two fragments: C4a (small fragment) and C4b (large fragment). C1 then proceeds to cleave the C2 component of the complement into C2a (large fragment) and C2b (small fragment). There are glycoproteins on the cell membrane with which C4b binds first, and C4b then binds to C2a to form the C4bC2a complex, which is called C3 convertase. C3 convertase in turn cleaves C3 into C3a (small fragment) and C3b (large fragment). C3a, being an anaphylotoxin, binds to C3a receptors on the membrane of mast cells and promotes their degranulation.

C3b binds to the free-circulating C5 component of complement. This results in conformational changes in C5, so that it becomes susceptible to the C4bC2a complex, which splits it into C5a (small fragment) and C5b (large fragment). C5a is also an anaphylotoxin and binds to C5a receptors on the membrane of mast cells, contributing to their degranulation [86,87].

C5b binds to C6, forming the C5bC6 complex. Further, C7 and then C8 attach to this complex, forming the C5bC6C7C8 complex. Eventually, this complex is incorporated into the cell membrane and 10 to 18 C9 proteins attach to it, polymerizing with each other and forming a pore on the cell membrane. This whole structure is called MAC (Membrane Attack Complex). The formation of the pore on the cell membrane leads to cell death [86,87].

#### 3.3.2. Alternative Pathway

It is important to mention that the C3 component of complement is an unstable molecule and spontaneously decomposes to C3a and C3b in the resting state in the blood plasma. However, only a small amount of C3 undergoes this reaction [86,87].

C3b binds to glycoproteins on the cell membrane. After that, C3b is joined by B protein. Then, under the action of factor D, B protein is split into two fragments—Bb (a large fragment), which remains bound to C3b, and Ba (a small fragment), which is released into the extracellular space. As a result, factor P (properdin) is attached to the C3bBb complex, which stabilizes this complex. The C3bBbP complex is also called the C3 convertase of the alternative pathway, which in turn begins to break down other C3 molecules in the blood plasma. This produces a large amount of C3b, which is a “+” feedback mechanism. Next, another C3b molecule joins the C3bBbP complex to form the C3bBbPBb—C5 convertase of the alternative pathway. Further steps are similar to those of the classical pathway [86,87].

#### 3.3.3. Lectin Pathway

In contrast to the classical complement pathway, lectin pathway is not initiated by antigen–antibody complex. Initiation of lectin pathway starts with binding of mannose-binding lectin (MBL) with terminal carbohydrate residues of D-mannose and other sugars with 3- and 4-OH groups placed in the equatorial plane, that can be found on the surfaces of certain pathogens. The function of MBL in lectin pathway is recognition of pathogen-associated carbohydrate patterns and activation of complement system [86,87].

Multimers of MBL form a complex with MBL-Associated Serine Proteases (MASP1 and MASP2), that are very similar to C1r and C1s molecules of the classical complement pathway. MASPs are protease zymogens. When the carbohydrate-recognizing heads of MBL bind to specifically arranged sugar residues on the surface of a pathogen, MASP1 and MASP2 start to cleave complement components C4 and C2. Cleavaged C4b binds to a bacterial cell, combines with C2a, and forms classical C3 convertase (C4bC2a) [86,87].

### 3.4. The Role of Histamine Liberators

Another important mechanism in the pathogenesis of SFP is the influence of histamine liberators on the mast cells. These substances induce the release of endogenous histamine from the secretory cells. In recent decades, many histamine-liberating amines, amides, guanines, guanidines, and other organic compounds have been discovered. Research regarding such substances has been conducted up to now, so the list of histamine liberators is constantly expanding [88].

It is noteworthy that mast cells may be activated directly by substances with low molecular mass—histamine liberators. Their activation is usually associated with systemic pseudoallergenic or peptide food supplements (the latter are usually given parenterally because digestive enzymes destroy them when given orally). Some molecules, for instance compound 48/80 (a polymeric amine, product of condensation between formaldehyde and p-methoxy-phenethyl-methylamine), may stimulate the activity of phospholipase D that activates synthesis of endogenous phospholipids which bind with the lysophosphatide acid receptor [89]. This process facilitates the activation of G-proteins, which may induce the release of mediators from the mast cells, synthesis of phosphatidylinositol-3-kinase, and the formation of arachidonate metabolites. Phosphatidylinositol-3-kinase stimulator activates phospholipase C gamma that stimulates phosphatidylinositol-4,5-bisphosphate hydrolysis that results in the formation of inositol-1,4,5-triphosphate and diacylglycerol [90]. These mediators induce the elevation of intracellular Ca^2+^ levels, which activates protein kinase C. This series of reactions leads to degranulation of the mast cells and the release of inflammatory mediators that are generated in the PTK-PLA2 pathway. These events lead to histamine leaving the cell and the development of the FPA [91].

It is assumed that cis-urocanic acid is the histamine liberator in the mechanism of SFP [92,93,94,95]. Cis-urocanic acid is a degradation product of the protein amino acid L-histidine. Deaminated histidine is converted to the more soluble cis isomer by UV [96]. Based on the assumptions of Wille et al., urocanic acid is a mast cell degranulator or histamine liberator [92]. Norval et al.’s studies have shown that the administration of cis-urocanoic acid, regardless of the route of administration, induces histamine release [97,98]. The structure of cis-urocanoic acid has been found to be similar to that of serotonin, having the property of binding to the 5-HT_2a_ receptor and activating immune suppression [99,100]. The ability of cis-urocanoic acid to induce mast cell degranulation has been demonstrated in human skin cultures, and depletion of the content of mast cell granules has been recorded [96].

Cis-urocanic acid in the pathogenesis of SFP is formed from the histidine of spoiled fish. Under the action of the putrefactive bacterial enzyme histidine ammonia-lyase (histidase), histidine is converted into trans-urocanic acid, which accumulates in fish [101,102]. Over time, the isomerization of trans-urocanic acid to cis-urocanic acid occurs. This process amplifies the ultraviolet radiation with its peak at 345 nm, e.g., storing fish in the sun [103,104,105]. It means that the concentration of histamine in SFP is elevated not only because of the exogenous histamine of the fish but also because of the endogenous histamine released from the mast cells [93] (Figure 6).

However, Zara D. et al. found that, in the case of tuna fish, the key link in the pathogenesis is histamine and not cis-urocanoic acid. When storing tuna fish at a temperature of 3 °C for 15 days, the level of histamine increased, while the level of cis-urocanoic acid remained low [106]. It can be concluded that fish storage temperature and exposure to UV are important in the mechanism of cis-urocanoic acid formation. Due to the insufficient study of trans- and cis-urocanoic acids in the food industry and the consequences of consuming cis-urocanoic acid exclusively for the human body, the limits of permissible concentration for this compound have not yet been established.

### 3.5. The Role of Associated Diseases and Disorders in the SFP Mechanism

Chronic gastrointestinal diseases, such as irritable bowel syndrome (IBS), inflammatory bowel disease (IBD), and gastroesophageal reflux disease (GERD), may disrupt the function of the intestinal mucosa and increase the absorption of histamine liberators into the mast cells of the mucosa. Because of that, these diseases play an important role in the development of FPA, such as SFP [56]. In that case, SFP is associated with individual sensitivity to products with a high content of cis-urocanic acid rather than a high amount of exogenous histamine and other biogenic amines in food products [93].

Another suggested pathological mechanism of SFP to a large amount of consumed exogenous histamine is the disruption of function of the catabolic enzyme DAO (DAO defect) in some people [71,107] (Figure 5). Petersen in 2002 and Schwelberger in 2004 discovered populations of patients genetically predisposed to gastrointestinal diseases with mutations in the SNP gene that codes for DAO [108,109]. It is possible that DAO synthesized in the intestines may have a crucial role in the development of histamine intolerance and the susceptibility of certain people to SFP [71]. This means that histamine intolerance may develop because of an enzyme deficiency or abnormal functioning. This is still an assumption because there are no reliable studies that show that decreased DAO activity might be a reason for the reaction to the ingested histamine. Many factors must be taken into consideration. Firstly, there is an alternative pathway of histamine metabolism in the human body, methylation by histamine-N-methyltransferase. More than that, histamine levels in food products vary depending on the conditions of storage, timing, and method of processing.

The leading role in the pathogenesis of SFP belongs to exogenous histamine. Histamine, in addition to its systemic effect, which leads to the development of the main symptoms of SPF (flushing, swelling, itchy rash, hypotension, acute pulmonary edema, bronchoconstriction) has an effect on the immune system. Histamine stimulates the secretion of large amounts of IL-4 and IL-5 by Th2-helper cells. IL-4 stimulates the proliferation of B cells, while IL-5 promotes the differentiation of B cells into plasma cells. It is important to know that the excessive release of IL-4, which occurs under the influence of histamine, leads to the selective synthesis of IgE by plasma cells. In turn, IgE will lead to prolonged mast cell activation, which may underlie the pathogenesis of patients developing mast cell activation syndrome and intestinal mastocytosis [110,111,112,113]. 

The molecular mechanism by which mast cell activation occurs under the action of exogenous histamine is shown in Figure 7. The mast cell membrane contains FcεRI receptors, which consist of one α, one β and two γ subunits. The cytoplasmic tails of the β and γ subunits contain immunoreceptor tyrosine-based activation motifs (ITAMs) that become phosphorylated upon FcεRI aggregation. This phosphorylation is mediated primarily by the Src family tyrosine kinase Lyn. Further, the Src homology (SH2) domain of Syk protein binds to the inorganic phosphate of γ subunits, and the Lyn molecule binds to the inorganic phosphate of β subunits with the same SH2 domain. Lyn phosphorylates Syk and thereby activates its kinase enzymatic function [114,115]. The linker for activation of T cells (LAT) protein is built into the membrane of mast cells and is a target for Syk and Lyn molecules, which phosphorylate LAT on multiple tyrosine residues. The inorganic phosphates on LAT protein serve as docking sites for SH2 domains of GRB2-related adaptor protein (GADS), growth-factor receptor-bound protein 2 (GRB2), and phospholipase Cγ1 (PLCγ1) [114,115].

The SH3 domain of the GRB2 molecule is bound to the proline-rich region of the Son of Sevenless (SOS) protein, which belongs to guanine nucleotide exchange factors (GEFs). In the quiescent state in the cell cytoplasm, there is RAS protein bound to GDP—inactive state. SOS detaches the GDP molecule from RAS and attaches GTP in its place—active state. Then RAS-GTP goes to the RAF-1 protein (proto-oncogene serine/threonine-protein kinase) and activates it. In turn, RAF-1 phosphorylates and thereby activates the enzyme MEK (MAPK/ERK kinase). MEK phosphorylates the mitogen-activated protein kinase (MAPK) molecule by the OH group of serine/threonine (mitogen-activated protein kinase) and thereby activates it. Ultimately, MAPK phosphorylates and activates specific transcription factors that increase the expression of genes responsible for cytokine synthesis (such as TNF-α, IL-8, and MCP-1) [114,115]. The SH3 domain of the GADS protein binds to SH2- domain-containing leukocyte protein of 76 kDa (SLP76), which is an adaptor protein for the attachment and activation of VAV. In turn, VAV being GEF performs the same function as SOS—MAPK activation [114,115].

PLCγ1 activates the inositol trisphosphate system by increasing the concentration of calcium in the cytoplasm and activating PKC. Calcium promotes the fusion of the vesicle membrane containing biologically active substances (BAS) with the cell membrane. This occurs by the following mechanism: the Soluble NSF attachment receptor (vSNARE) proteins Synaptobrevin and Synaptotagmin are on the vesicle membrane, and the tSNARE proteins Syntaxin and SNAP-25 are on the mast cell membrane. Under the influence of calcium, the vSNARE proteins intertwine with the tSNARE proteins, bringing the vesicle membrane closer to the cell membrane and eventually promoting their fusion. That is, vesicle exocytosis occurs and BAS enters the extracellular space. In addition, PKC itself phosphorylates the vesicles and thus promotes their fusion with the mast cell membrane. One of the molecules released during mast cell degranulation is histamine. This leads to a vicious circle and exacerbates the allergic reaction [114,115,116].

It is important to know that calcium ions and MAPK activate cPLA2α, increasing the production by mast cells of prostaglandins, thromboxanes, leukotrienes, lipoxins, resolvins, and eoxins. This occurs by the following mechanism: the molecule Cytoplasmic phospholipase A2α (cPLA2α) has a C2 domain and a catalytic domain. Two calcium cations bind to the C2 domain of cPLA2α, stimulating the interaction of this domain with phosphatidylcholine (PC) located on the perinuclear membrane. Thus, cPLA2α gains access to the phospholipids of the perinuclear membrane, hydrolyzing them to form arachidonic acid. But calcium alone is not sufficient for prolonged activation of cPLA2α. MAPK phosphorylates cPLA2α by the OH group of serine/threonine, which changes the conformation of this molecule so that the catalytic domain of cPLA2α binds to Phosphatidylinositol-4,5-bisphosphate (PIP2), a constituent of the perinuclear membrane, which ensures long-term binding of cPLA2α to the membrane, even in the absence of calcium ions [114,115].

As we discussed previously, plasma cells are forced to synthesize IgE under the influence of a huge amount of exogenous histamine. IgE activates mast cells that secrete many biological active mediators, which include β-Hex, HIS, and pro-inflammatory cytokines (such as TNF-α, IL-8, and MCP-1). All these molecules, and especially TNF-α, lead to the development of the inflammation, mainly through the activation of NF-κB pathway [117]. There are two main NF-κB pathways: canonical and non-canonical (Figure 8).

#### 3.5.1. Canonical NF-κB Pathway

In the quiescent state, TNFR1 (tumor necrosis factor receptor) is embedded in the cell membrane as monomers. The cytoplasmic part of TNFR1 contains DD (Death Domain), to which the molecule SODD (Silence Of Death Domain) is linked, blocking the activity of DD. When TNF-α binds to TNFR1, TNFR1 undergoes homotrimerization. In this state, TNFR1 changes its conformation, which leads to the detachment of SODD from the DD of TNFR1. Further, the adaptor protein TRADD (Tumor necrosis factor receptor type 1-associated DEATH domain protein) binds to DD. In turn, TRADD recruits the molecule RIPK1 (Receptor-interacting serine/threonine-protein kinase 1) and TRAF2 (TNF receptor-associated factor 2). TRAF2 then recruits the molecule IKK (IκB kinase), which consists of three subunits: IKK-α, IKK-β, and IKK-γ (NEMO—NF-kappa-B essential modulator). RIPK1 phosphorylates IKK-β by the OH-groups of the serine/threonine residues, leading to activation of the IKK molecule [118,119].

The cell cytosol contains the NF-κb molecule, which consists of three subunits: IkBα (Inhibitor of Nuclear Factor (NF)-Κb α isoform), p50, and RelA. In the resting state, IkBα masks the NLS (nuclear localization sequence) of p50 and RelA proteins, preventing transport of these subunits as a heterodimer from the cell cytosol into the nucleoplasm. However, upon activation, IKK starts phosphorylating the IkBα subunit by the OH-groups of the serine/threonine residues. Further, the enzyme E3 ubiquitin ligase binds to the phosphorylated serine/threonine residues, promoting ubiquitination and further proteosomal degradation of IkBα. As a result, NLS is unmasked and the p50/RelA heterodimer is transferred from the cell cytoplasm into the nucleoplasm [119,120].

#### 3.5.2. Non-Canonical NF-κB Pathway

In the quiescent state of the cell, cytosol contains TRAF2 molecule, to which cIAP (Cellular inhibitor of apoptosis protein 1) and TRAF3 molecules are bound. In turn, the NIK (NF-kappa-B-inducing kinase) molecule is linked to TRAF3. cIAP performs ubiquitination of the NIK, contributing to its proteosomal degradation. This suggests that in the resting state, the concentration of NIK in the cell cytosol is very low [118,120].

TRAF2 in complex with TRAF3 and cIAP are recruited to the receptor. NIK dissociates from this complex and is free in the cytoplasm of the cell. Since NIK is free, cIAP cannot ubiquitinate it, resulting in an increase in the concentration of NIK in the cell cytosol. Recruitment of the TRAF2/TRAF3/cIAP complex to the receptor causes cIAP to begin to ubiquitinate TRAF3 within this complex, contributing to its proteosomal degradation. This leads to a decrease in the concentration of TRAF3 in the cell cytoplasm, thus preventing the assembly of a new TRAF2/TRAF3/cIAP/NIK complex, resulting in the inability to ubiquitinate NIK and, consequently, an increase in NIK concentration [119,121]. 

The cell cytosol contains the NF-κB molecule, which consists of two subunits: p100 and RelB. However, this is the inactive state of NF-κB. NIK starts phosphorylating the IKKα/IKKα homodimer by the OH-groups of the serine/threonine residues, leading to its activation. It is important to note that this complex exists without NEMO protein. In turn, IKKα, on the one hand, phosphorylates NIK, which leads to its ubiquitination and further complete proteosomal degradation (negative feedback mechanism), and on the other hand, phosphorylates p100, which leads to its ubiquitination and partial proteosomal degradation. As a result, p100 is converted into p52 and the p52/RelB complex is formed, which is transported into the nucleoplasm [120,121].

Both p52/RelB and p50/RelA complexes are transcription factors that increase the expression of the genes responsible for the synthesis of cytokines (TNF-α, IFN-β, IL-β, IL-2, IL-3, IL-6, and etc.), chemokines (IL-8, RANTES, MCP-1), adhesion molecules (E-selectin, ICAM-1, and VCAM-1), acute response protein, complement, growth factor (GM-CSF, G-CSF, and M-CSF), etc. [119].

## 4. Key Points in the Relief of the SFP

Despite the specificity of the molecular mechanisms of SFP, treatment approaches remain extremely similar, as in the management of IgE-associated food allergies. The following suggestions for managing patients with SFP should be taken into consideration. The patients may have both a slow development of symptoms and an acute current, such as an anaphylactoid reaction. This will determine the tactics to be used. In either case, the trigger—fish and fish products—must be eliminated. 

In case of an anaphylactoid reaction (with bronchospasm, airway edema, or distributive shock), the patients should be treated for anaphylaxis with epinephrine (adrenaline) and methylprednisolone [4,32,61]. In case of hypotension, low doses of vasopressors can be added to therapy [61].

In most cases, treatment is supportive. In case of mild severity of symptoms, the effective approach to treatment is an immediate administration of antihistamines for 1 or 2 days. H1-Antihistamines are the mainstay of treatment [122]. For mild to moderate symptoms, oral H1 histamine antagonists such as diphenhydramine, cetirizine, and chlorpheniramine are effective [61]. H2 receptor antagonists (cimetidine, famotidine, or ranitidine) may also be added to therapy [61]. With this treatment, symptoms should resolve in 6–8 h [32]. However, there are no trials with quality control that can validate this recommendation or preference of one histamine antagonist or combination over others [61].

Intravenous administration of antihistamines, such as diphenhydramine, famotidine, and ranitidine, can be performed if patient does not tolerate oral antihistamines. Nausea and vomiting can be treated with intravenous promethazine. For patients experiencing dehydration, intravenous fluid should be administrated [61].

According to the symptoms, beta-2-adrenergic agonists, ipratropium bromide, and steroids may also be prescribed [4]. In addition, there is insufficient data to support the prophylactic use of mast cell stabilizers [123]. Another focus of attention for us should be the comorbidities that play a role in the SFP mechanism. The optimal treatment consists of a multidisciplinary and multifaceted approach, which encompasses long-term management strategies in order to minimize recurrences of reactions and improve quality of life [124].

## 5. Conclusions

The molecular mechanisms of SFP include a complex system of interactions between the body and chemical triggers such as exogenous histamine, other biogenic amines, and cis-urocanic acid. The onset and development of the molecular mechanism of SFP are influenced by external contributing factors (fish species, saprophytic microorganism species that convert histidine to histamine, and environmental conditions) and internal individual factors (exogenous histamine consumed, inactivation speed, individual sensitivity, etc.). The main roles in the molecular mechanisms of SFP are not only dose-dependent exogenous histamine intoxication but also the complement system, NET activation, NF-κB, etc. Important in the development of SFP are associated diseases and disorders, such as IBS, IBD, GERD, mast cell activation syndrome, intestinal mastocytosis, and others. Therefore, SFP is a systemic process realized by various mechanisms that still offers many challenges for hygienic regulations because aspects of the identification of maximum allowable concentrations of some substances and their influence on the pathogenesis of this disease remain unclear.

## Figures and Tables

**Figure 1 ijms-24-00809-f001:**
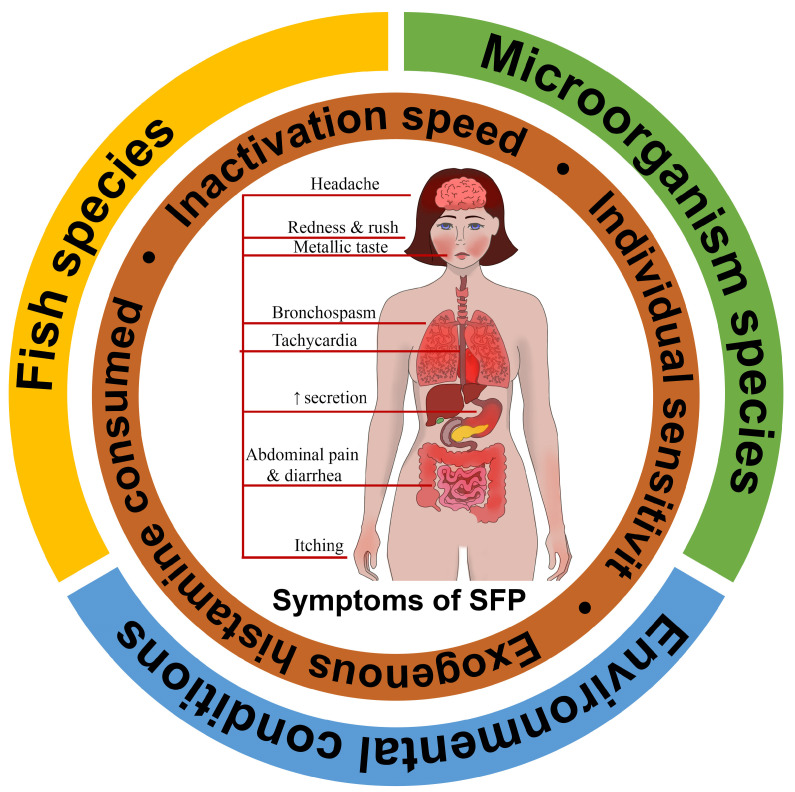
Scombroid food poisoning formula. External contributing factors: fish species, saprophytic microorganism species that convert histidine to histamine, and environmental conditions; Internal individual factors: exogenous histamine consumed, inactivation speed, and individual sensitivity, etc.

**Figure 2 ijms-24-00809-f002:**
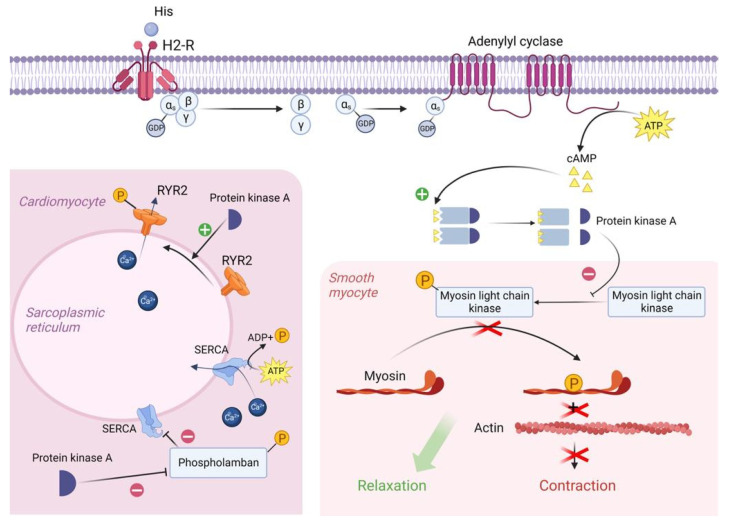
Molecular mechanism of H2 receptor activation by exogenous histamine in SFP. Exogenous histamine binds to the H2-R and leads to the adenylyl cyclase system activation. In smooth muscle cells, PKA phosphorylates MLCK, leading to the smooth muscle cell relaxation. In cardiomyocytes, PKA phosphorylates Phospholamban and RYR2, resulting in an increased heart rate. PKA—proteinkinase A, GTP—guanosine triphosphate, MLCK—myosin light chain kinase, GDP—guanosine diphosphate, cAMP—cyclic adenosine monophosphate, RYR2—ryanodine receptor 2, SR—sarcoplasmic reticulum, SERCA—sarco/endoplasmic reticulum Ca—ATPase.

**Figure 3 ijms-24-00809-f003:**
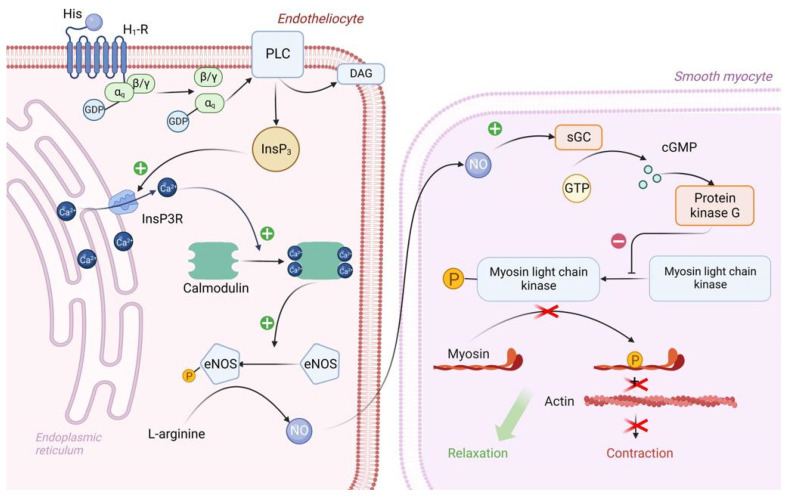
Molecular mechanism of H1 receptor activation by exogenous histamine in SFP. Exogenous histamine binds to the endothelial H1-R and leads to the inositol phosphate pathway activation. InsP3 stimulates Ca release, thereby leading to the Ca/Calmodulin complex formation. Ca/Calmodulin complex activates eNOS, which catalases NO formation. NO enters the sarcoplasm of smooth muscle cells and eventually activates PKG. PKG phosphorylates MLCK, leading to smooth muscle cell relaxation. PLC—Phospholipase C, PIP2—phosphotidylinositol diphosphate, InsP3—inositol triphosphate, DAG—diacylglycerol, PKC—proteinkinase C, eNOS—endothelial NO synthase, sGC—soluble guanylate cyclase, GTP—guanosine diphosphate, cGMP—cyclic guanosine monophosphate, PKG—proteinkinase G, MLCK—myosin light chain kinase.

**Figure 4 ijms-24-00809-f004:**
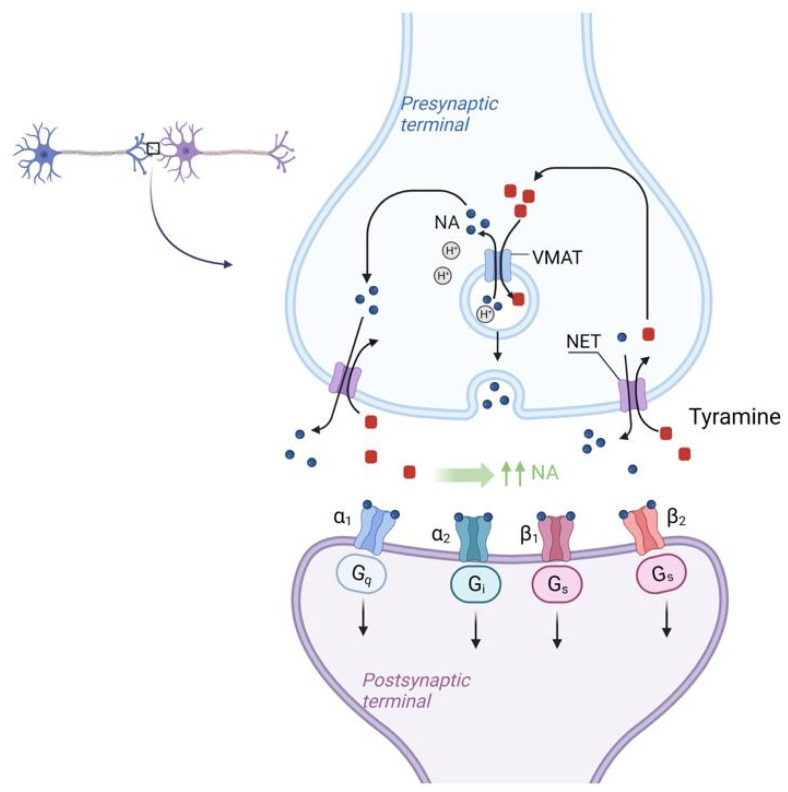
Molecular mechanism of NET (Norepinephrine transporter) activation by tyramine in SFP. Tyramine from the synaptic cleft is transported to the synaptic bulb by the NET (Norepinephrine transporter) in exchange for NE itself. Once inside the synaptic bulb, tyramine is transferred inside the vesicle by the vMAT (vesicular monoamine transporter), located on the membrane of these vesicles, in exchange for NE. This NE then enters the synaptic cleft again via the NET in exchange for tyramine and the cycle repeats. As a result, there is a sharp increase in the concentration of NE in the synaptic cleft. NET—norepinephrine transporter, vMAT—vesicular monoamine transporter.

**Figure 5 ijms-24-00809-f005:**
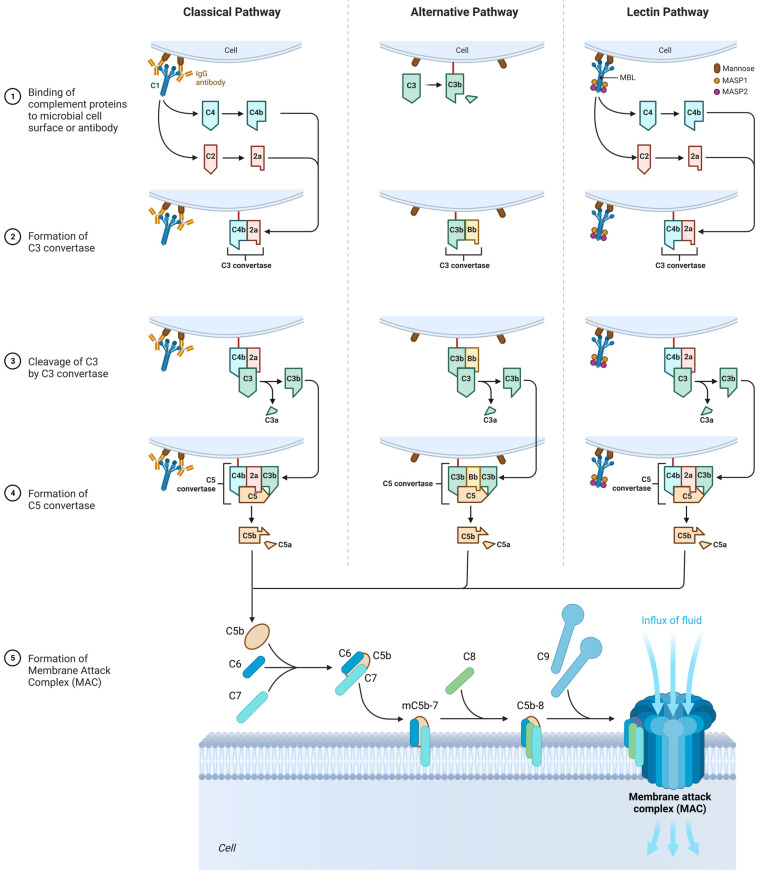
The role of the complement system in the SFP mechanism. Classical pathway: C1 component of complement cleaves C4 and C2 into two fragments. C3 convertase is formed. Under the action of C3 convertase, C3a will be formed. C3a, being an anaphylotoxin, will lead to the basophil degranulation. At the end, MAC is formed that leads to cellular damage. Alternative pathway: C3 spontaneously decomposes to C3a and C3b. After that, C3bBbP complex is formed (C3 convertase of the alternative pathway), and more C3 decomposes to C3a and C3b. The end step is MAC formation. Lectin pathway: All the characteristics are similar to those in classical pathway, except initiation factor. In this case, initiation factor is MBL.

**Figure 6 ijms-24-00809-f006:**
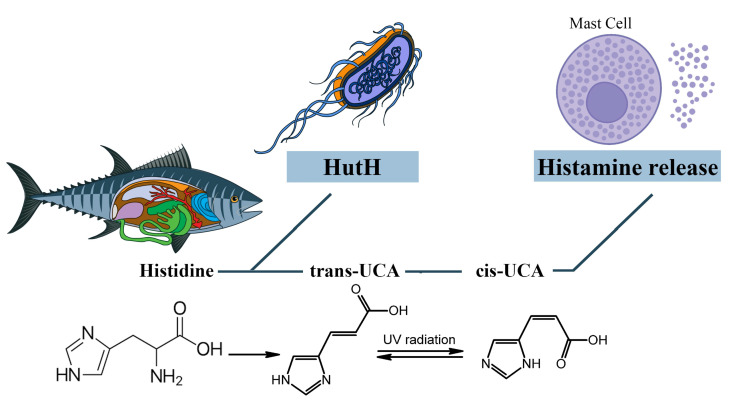
The role of cis-urocanic acid in the SFP mechanism. Histidine is converted into trans-urocanic acid, which accumulates in fish. Over time, the isomerization of trans-urocanic acid to cis-urocanic acid occurs, which plays the role of histamine liberator. HutH—histidine ammonia-lyase, UCA—urocanic acid.

**Figure 7 ijms-24-00809-f007:**
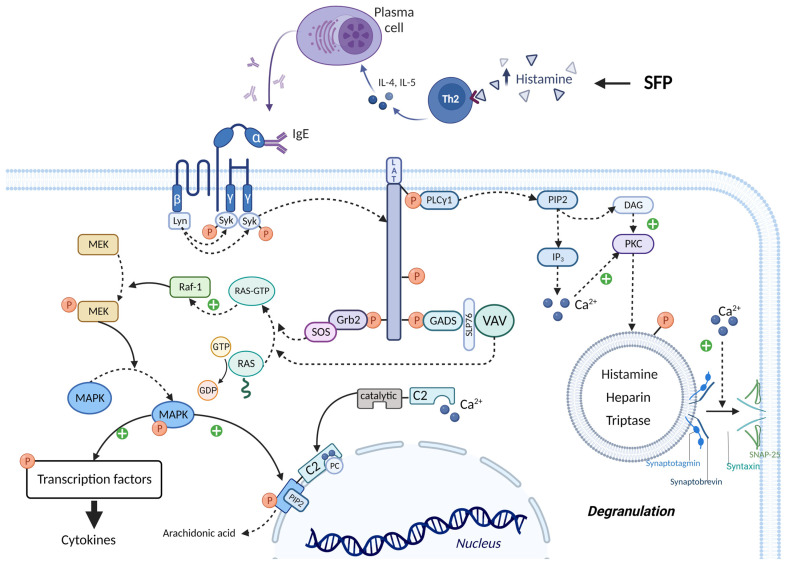
IgE-mediated activation of mast cells in SFP. IgE binds to the FcεRI receptors and leads to the activation of Lyn and Syn. Syn phosphorylates LAT, thereby creating a platform for recruitment of Grb2, GADS, and PLCγ1. Grb2 and GADS activate biochemical cascades which lead to MAPK activation and cytokines production. PLCγ1 activates IP_3_ system and leads to the mast cell degranulation and arachidonic acid production. ITAMs—immunoreceptor tyrosine-based activation motifs, SH2—Src homology domain, LAT—linker for activation of T cells, GADS—GRB2-related adaptor protein, GRB2—growth-factor receptor-bound protein 2, PLCγ1—phospholipase Cγ1, SOS—the Son of Sevenless, GEFs—guanine nucleotide exchange factors, RAF-1—proto-oncogene serine/threonine-protein kinase, MEK—MAPK/ERK kinase, MAPK—mitogen-activated protein kinase, SLP76—SH2-domain-containing leukocyte protein of 76 kDa, cPLA2α—cytoplasmic phospholipase A2α.

**Figure 8 ijms-24-00809-f008:**
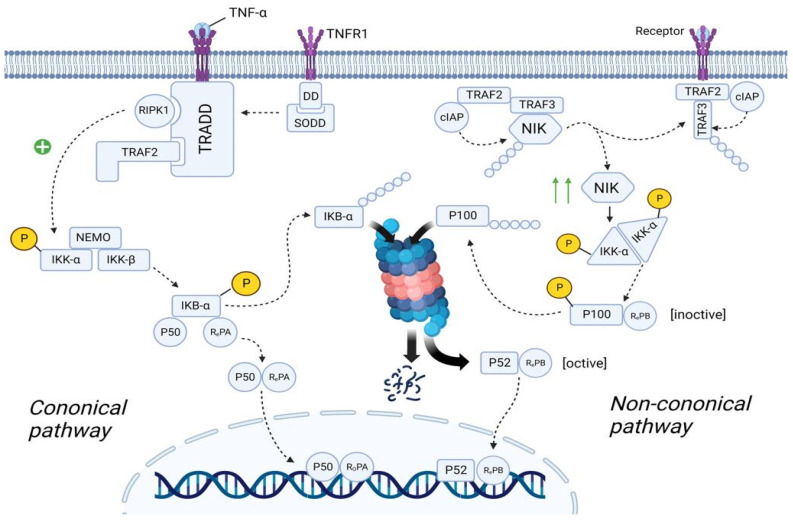
The role of the NF-κB in the SFP mechanism. Canonical pathway: TNF-α binds to TNFR1 and TNFR1 undergoes homotrimerization. Then, SODD dissociates from the TNFR1, which will lead to the recruitment of TRADD. In turn, TRADD recruits RIPK1 and IKK. RIPK1 phosphorylates and activates IKK. IKK starts phosphorylating the IkBα subunit of the NF-kβ and thereby promotes ubiquitination and further proteosomal degradation of IkBα. As a result, NF-kβ is transported to the nucleus. Non-canonical pathway: In the resting state, TRAF2/TRAF3/cIAP/NIK complex exists. Formation of this complex will lead to the proteosomal degradation of NIK. After ligand–receptor interaction, TRAF2/TRAF3/cIAP complex dissociates from NIK, preventing its proteosomal degradation. The number of NIK in the cell increases. NIK starts phosphorylating and activating the IKKα. IKKα leads to the formation of the p52/RelB complex and its translocation to the nucleus. TNFα—tumor necrosis factor α, TNFR1—tumor necrosis factor receptor1, DD—death domain, SODD—silence of death domain, TRADD—tumor necrosis factor receptor type 1-associated death domain protein, RIPK1—receptor-interacting serine/threonine-protein kinase 1, TRAF2—TNF receptor-associated factor 2, IKK—IκB kinase, NEMO—NF-kappa-B essential modulator, IkBα—inhibitor of nuclear factor (NF)-Κb α isoform, NLS—nuclear localization sequence, cIAP—cellular inhibitor of apoptosis protein 1, NIK—NF-kappa-B-inducing kinase.

**Table 1 ijms-24-00809-t001:** Bacteria species that can produce histamine and their characteristics.

Bacteria	Conditions of Histamine Formation/ Food Source	Level of Histamine Forming	Reference
*Morganella morganii*	15–37 °C, pH < 7/Fish, Tuna Salad	>5000 ppm	[34,37]
*Photobacterium phosphoreum*	4–37 °C, pH < 7/Fish	1188 ppm	[34]
*Photobacterium kishitanii*	20–37 °C, pH < 7/Fish	1545 ppm	[34]
*Klebsiella pneumoniae*	37 °C, pH < 7/Fish, Swiss cheese	442 ppm	[34,38,39]
*Raoultella planticola* (Synonym: *Klebsiella* *pneumoniae strain T2* or *Klebsiella planticola* (ATCC 43176))	20–37 °C, pH < 7/Fish	between 2810 and 5250 mg/L	[38]
*Raoultella ornithinolytica*	20–37 °C, pH < 7/Fish	between 2810 and 5250 mg/L	[38]
*Clostridium perfringens*	20–37 °C, pH < 7/Fish	19 ppm in tuna, 3 ppm in spanish mackerel	[40,41]
*Hafnia alvei*	30–37 °C, pH < 7/Fish, Fish broth	>88.7 ppm (30 °C), 42.1 ppm (15 °C)	[42,43]
*Enterobacter cloacae*	30–37 °C, pH < 7/Fish broth	>1000 ppm	[42]
*Citrobacter freundii*	30–37 °C, pH < 7/Fish	>1600 ppm (37 °C), 474 ppm (30 °C)	[30,43,44]
*Escherichia coli*	30–37 °C, pH < 7/Fish	Not detected <1 ppm, but they have the enzyme	[30,43]

**Table 2 ijms-24-00809-t002:** The distinctions in mechanisms between true IgE-associated allergies to fish and seafood and SFP.

Characteristic	IgE-Associated Allergy to Fish and Seafood	Scombroid Food Poisoning
*Common mechanisms*		
Histamine release	Yes	
Activation of the complement system	Yes	
Atypical eicosanoid synthesis	Yes	
Inhibiting of bradykinin decomposition	Yes	
*Different mechanisms*		
Dose dependence on antigen/allergen	Occasionally, depends on the molecular structure of the antigen	Always
Hidden sensibilization to the antigen	Yes	No
Elevation of non-specific IgE in serum	Often	Occasionally
Elevation of specific IgE	Always	Never
Concentration of the substance that induces the reaction	Low	High
Immunological stage	Yes	No
Formation of antigen-specific immune complexes	Yes	No
Influence on the mast cells	IgE-mediated influence	Direct influence with the substance
Mediators	Endogenous histamine, tryptase	Exogenous histamine, histamine liberators, serotonin liberators
DAO defect	Rare	Possible

## Data Availability

The data presented in this study are available on request from the corresponding author.

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
