# Peer review of "Molecular Mechanisms of Scombroid Food Poisoning"

_ijms, 2023, doi:10.3390/ijms24010809_

Round 1
Reviewer 1 Report
The manuscript describes the mechanisms of scombroid food poisoning in details. The authors tried to fully covered all issues linked with the main topic. Good supportive figures are included. Few small changes are suggested to be made:
Abstract:
1. Please start the abstract from explanation of SFP first and then mention that SFP is a part of FPA. FPA means Food Pseudo Allergic, please correct "food pseudo-allergic allergy" into food pseudo allergic reactions.
2. Line 26: Too long sentence: "The most important difference is the absence of IgE in 26 SFP. Because of the great variety of molecular pathways in this process, it is very important to un- 27 derstand the differences between them for a conscious approach to prophylaxis of SFP induced by 28 fish products may prevent poor outcomes in potential patients and help to develop an adequate 29 system of hygienic regulations to control the amount of substances in food products that may in- 30 duce FPA." Please rewrite.
3. Line 33. This sentence: "Despite the 33 vast and lengthy history of research on SFP mechanisms, there are still many blank spots in our 34 understanding of this condition. Pathogenesis includes a complex system of interactions between 35 the body and chemical triggers such as exogenous histamine, other biogenic amines, cis-urocanic 36 acid, salicylates, and other histamine liberators. " should be placed somewhere before the mentioned goals in previous sentence in abstract.
4. From line 37: "In this overview, you will learn about the roles of 37 the complement system, NET activation, and NF-kB in the SFP mechanisms. So, SFP is a systemic 38 process realized by various mechanisms that still offers many challenges for hygienic regulations 39 because aspects of the identification of maximum allowable concentrations of some substances and 40 their influence on the pathogenesis of this disease remain unclear." 'till the end. This part should be deleted from abstract.
INTRODUCTION
1. Line 46: FPA goes with pseudo-allergic reactions to food. In different way false food allergy (FFA). Please reconsider the abbreviation used.
2. Line 48: Non-immunologic food intolerance does not include FPA. The Reviewer suggest to write" Non-immunological adverse reactions to food including FPA...".
----------------
Bibliography style needs to unified.
Author Response
|
Reviewer 1 The manuscript describes the mechanisms of scombroid food poisoning in details. The authors tried to fully covered all issues linked with the main topic. Good supportive figures are included. Few small changes are suggested to be made: Abstract: 1. Please start the abstract from explanation of SFP first and then mention that SFP is a part of FPA. FPA means Food Pseudo Allergic, please correct "food pseudo-allergic allergy" into food pseudo allergic reactions. |
Dear Reviewer 1, Thank you very much for your advice and correction! We greatly appreciate your time spent reviewing our manuscript. Annotation corrected; necessary changes made (Lines 23-25).
|
|
2. Line 26: Too long sentence: "The most important difference is the absence of IgE in 26 SFP. Because of the great variety of molecular pathways in this process, it is very important to un- 27 derstand the differences between them for a conscious approach to prophylaxis of SFP induced by 28 fish products may prevent poor outcomes in potential patients and help to develop an adequate 29 system of hygienic regulations to control the amount of substances in food products that may in- 30 duce FPA." Please rewrite. |
The sentence in the abstract has been rewritten (Line 30). |
|
3. Line 33. This sentence: "Despite the 33 vast and lengthy history of research on SFP mechanisms, there are still many blank spots in our 34 understanding of this condition. Pathogenesis includes a complex system of interactions between 35 the body and chemical triggers such as exogenous histamine, other biogenic amines, cis-urocanic 36 acid, salicylates, and other histamine liberators. " should be placed somewhere before the mentioned goals in previous sentence in abstract. |
The sentence has been moved (Line 36). |
|
4. From line 37: "In this overview, you will learn about the roles of 37 the complement system, NET activation, and NF-kB in the SFP mechanisms. So, SFP is a systemic 38 process realized by various mechanisms that still offers many challenges for hygienic regulations 39 because aspects of the identification of maximum allowable concentrations of some substances and 40 their influence on the pathogenesis of this disease remain unclear." 'till the end. This part should be deleted from abstract. |
This part has been deleted from the abstract. |
|
INTRODUCTION 1. Line 46: FPA goes with pseudo-allergic reactions to food. In different way false food allergy (FFA). Please reconsider the abbreviation used. |
Thank you for your comment! We have added text and clarified the abbreviation FFA and FPA (Lines 45-47). |
|
2. Line 48: Non-immunologic food intolerance does not include FPA. The Reviewer suggest to write" Non-immunological adverse reactions to food including FPA...". |
Reviewer agreement. Changes have been made (Line 48). |
|
Bibliography style needs to unified. |
Corrected |

Reviewer 2 Report
Even though this review is quite comprehensive and well written there are many issues that need to be addressed starting with the title
Major issues
1. The review does NOT discuss relevant molecular mechanisms, especially any current advances and is therefore misleading.
2. The term "thermostable histamine" is used but there is no explanation what it means or how it is stable since bigenic amines can be easily oxidized.
3. Section 3.5 There is no evidence presented for what molecules in fish may act as mast cell liberators. An example is mentioned of cis-urocanic acid, but there are no references provided. hence, this section should be expanded or minimized and Fig. 5 deleted.
4. What are "scombroid toxins"?
5. Sections 3.2 and 3.3 The notation Fig. 3 is used twice for two different topics and with two different legends. In fact both of these figs and topics are irrelevant because they do not directly address scombroid toxicity but discuss release of amines and signal-transduction pathways of TNFD etc
6. Section 3.6 The discussion of salicylates is confusing. Sensitivity to salicylates in astmatic is differet and not obvious how it contributes to scombroid toxicity.
7. Section 3.7 is inadequate as the is only mention of IBS and no discussion of mast cell activation diseases including
Mast cell activations syndrome
Intestinal mastocytosis
Cyclic-vomiting syndrome
Hsieh FH. Gastrointestinal Involvement in Mast Cell Activation Disorders. Immunol Allergy Clin North Am. 2018 Aug;38(3):429-441. doi: 10.1016/j.iac.2018.04.008. PMID: 30007461.
especially the ability of mast cells to secrete mediators without histamine.
Theoharides TC, Leeman SE. Effect of IL-33 on de novo synthesized mediators from human mast cells. J Allergy Clin Immunol. 2019 Jan;143(1):451. doi: 10.1016/j.jaci.2018.09.014. Epub 2018 Nov 1. PMID: 30390921.
8. Section 4 Recommendations does not discuss either acute interventions such as administration of anti-histamines, mast cell stabilizers, sysemic support etc, hence its clinical usefulness is quite limited.
Author Response
|
Reviewer 2 Even though this review is quite comprehensive and well written there are many issues that need to be addressed starting with the title
Major issues
1. The review does NOT discuss relevant molecular mechanisms, especially any current advances and is therefore misleading.
|
Dear Reviewer 2, Many thanks for the in-depth analysis and review of our manuscript! We greatly appreciate your time. We have changed the title of the manuscript to "Molecular Mechanisms of Scombroid Food Poisoning." We have also significantly revised and restructured our manuscript in accordance with your comments. We hope that in this form it will be of interest to the readers of the IJMS MDPI! |
|
2. The term "thermostable histamine" is used but there is no explanation what it means or how it is stable since bigenic amines can be easily oxidized. |
Thank you for your comment! We removed the phrase "thermostable histamine" from the text. It meant histamine, which is formed by the action of heating (cooking) fish. We gave an explanation, according to the text of the manuscript, in Section 2.3. «Environmental conditions» about the possibility of the formation of histamine when heated (pls. see Lines 175-179).
|
|
3. Section 3.5 There is no evidence presented for what molecules in fish may act as mast cell liberators. An example is mentioned of cis-urocanic acid, but there are no references provided. hence, this section should be expanded or minimized and Fig. 5 deleted. |
Reviewer agreement. New links have been added, and section 3.4 has been expanded (Lines 501-511). Figure 6 has been redrawn. |
|
4. What are "scombroid toxins"? |
Thank you for pointing out the incorrect term! This term has been removed from the review. "Scombroid toxins" was the historical name of histamine and histamine-like substances. |
|
5. Sections 3.2 and 3.3 The notation Fig. 3 is used twice for two different topics and with two different legends. In fact both of these figs and topics are irrelevant because they do not directly address scombroid toxicity but discuss release of amines and signal-transduction pathways of TNFD etc |
The typo in the notation of the figure has been corrected. We have added information about the association of tyramine pathways with SFP (Lines 401-403) and signal-transduction pathways of TNF with SFP (section moved to Section 3.5. "The role of associated diseases and disorders in the SFP mechanism" Line 634). |
|
6. Section 3.6 The discussion of salicylates is confusing. Sensitivity to salicylates in astmatic is differet and not obvious how it contributes to scombroid toxicity. |
Thank you for your comment! It was decided to delete Salicylate's Section 3.6 completely.
|
|
7. Section 3.7 is inadequate as the is only mention of IBS and no discussion of mast cell activation diseases including
Mast cell activations syndrome
Intestinal mastocytosis
Cyclic-vomiting syndrome
Hsieh FH. Gastrointestinal Involvement in Mast Cell Activation Disorders. Immunol Allergy Clin North Am. 2018 Aug;38(3):429-441. doi: 10.1016/j.iac.2018.04.008. PMID: 30007461.
especially the ability of mast cells to secrete mediators without histamine.
Theoharides TC, Leeman SE. Effect of IL-33 on de novo synthesized mediators from human mast cells. J Allergy Clin Immunol. 2019 Jan;143(1):451. doi: 10.1016/j.jaci.2018.09.014. Epub 2018 Nov 1. PMID: 30390921. |
Thank you very much for marking the section we missed! We have added a discussion of mast cell activation diseases (pls. see Lines 557-633). Figure 7 has been added. In this section, we also demonstrated the significance of the previously described TNF signal-transduction pathways for scumbroyd poisoning (Lines 634-710). We believe it is critical to conclude this section with a detailed explanation of the molecular mechanism of SFP, demonstrating the activation pathway of mast cells under the influence of exogenous histamine with the release of TNF-alpha and activation of the NF-kB pathway. |
|
8. Section 4 Recommendations does not discuss either acute interventions such as administration of anti-histamines, mast cell stabilizers, sysemic support etc, hence its clinical usefulness is quite limited. |
Thank you for your comment! To increase the clinical usefulness of the review, we have written Section 4 «Key points in the relief of the SFP» separately (pls. see Lines 711-741). Recommendations for the sanitary standards of fish storage have been moved to the restructured Section 2 «Contributing factors of scombroid food poisoning» (Lines 84-107 and 158-198). A new Figure 1 has also been added. |
|
Open Review Section Is the work scientifically sound and not misleading? 2 stars out of 5
|
We have added data on the low awareness of medical personnel about the SFP in the introduction (Line 56-59). The section 4 «Key points in the relief of the SFP» (clinical significance) was also written separately (Lines 711-741). All this should increase the significance of our review. We hope that the restructuring of the manuscript and corrections after review should improve the quality of the material submitted. |
|
Are there appropriate and adequate references to related and previous work? 2 stars out of 5 |
This review is a continuation of another our review in a special issue «Molecular Mechanisms of Allergy and Asthma 2.0» of the IJMS MDPI: Zhernov, Y.V.; Vysochanskaya, S.O.; Sukhov, V.A.; Zaostrovtseva, O.K.; Gorshenin, D.S.; Sidorova, E.A.; Mitrokhin, O.V. Molecular Mechanisms of Eosinophilic Esophagitis. Int. J. Mol. Sci. 2021, 22, 13183. https://doi.org/10.3390/ijms222413183. We wanted to describe in detail the molecular mechanisms of food allergies and pseudo-allergic reactions for physicians and medical students. |
